



# Traceable total ozone column retrievals from direct solar spectral irradiance measurements in the ultraviolet

Luca Egli[1], Julian Gröbner[1], Gregor Hülsen[1], Herbert Schill[1] and René Stübi[2]

[1]Physikalisch-Meteorologisches Observatorium Davos (PMOD/WRC), 7260 Davos Dorf, Switzerland
[2]MeteoSwiss, Payerne, Switzerland

*Correspondence to*: L. Egli (luca.egli@pmodwrc.ch)

**Abstract** Total column ozone (TCO) is commonly measured by Brewer and Dobson spectroradiometers. Both types of instruments are using four wavelengths in the ultraviolet radiation range to derive TCO. For the calibration and quality assurance of the measured TCO both instrument types require periodic field comparisons with a reference instrument.

This study presents traceable TCO retrievals from direct solar spectral irradiance measurements with the portable UV reference instrument QASUME. TCO is retrieved by a spectral fitting technique derived by a minimal least square fit algorithm using spectral measurements in the wavelength range between 305 nm and 345 nm. The retrieval is based on an atmospheric model accounting for different atmospheric parameters such as effective ozone temperature, aerosol optical depth, Rayleigh scattering, $SO_2$, ground air pressure, ozone absorption cross sections and top-of-atmosphere solar spectrum. Traceability

means, that the QASUME instrument is fully characterized and calibrated in the laboratory to SI standards (International System of Units). The TCO retrieval method from this instrument is independent from any reference instrument and does not require periodic in situ field calibration.

The results show that TCO from QASUME can be retrieved with a relative standard uncertainty of less than 0.8%, when accounting for all possible uncertainties from the measurements and the retrieval model, such as different cross sections,

different reference solar spectra, uncertainties from effective ozone temperature or other atmospheric parameters. The long-term comparison of QASUME TCO with a Brewer and a Dobson in Davos, Switzerland, reveals, that all three instruments are consistent within 1% when using the ozone absorption cross section from the University of Bremen. From the results and method presented here, other absolute SI calibrated cost effective solar spectroradiometers, such as array spectroradiometers, may be applied for traceable TCO monitoring.

**1 Introduction**

Since the 1970's the depletion of the stratospheric ozone layer is reported in many international scientific publications (e.g. Molina and Rowland 1974, Solomon 1999, Staehlin et al. 2001) and regularly summarized in the WMO assessment of ozone depletion (https://www.esrl.noaa.gov/csd/assessments/ozone). It has been shown that the ozone layer in the stratosphere was reduced by anthropogenic emissions of ozone depleting substances, which have been controlled since the 1980's by the

Montreal protocol. In order to monitor the effect of the decisions by the Montreal protocol and its amendments, the temporal development of the ozone layer needs to be observed worldwide with accurate instrumentations. Furthermore, the mutual interactions between the ozone layer and climatic change on Earth surface and variations of the solar constant are currently investigated and require long-term consistent observations of total ozone column in the atmosphere (e.g. Bais et al., 2015, Bais et al., 2019; Seckmeyer et al., 2018, Young et al., 2021).

The atmospheric shield of total column ozone (TCO) is important for the incoming UV radiation at earth surface and its impact on human health (Zerefos, 2002) due to changing UV exposure on the ground with decreasing TCO. Since the ozone layer absorbs effectively the radiation in the UV band at wavelengths shorter than 350 nm, accurate direct sun measurements of the UV radiation allow retrieving TCO at the earth surface by measuring the UV radiation in that wavelength band (Kerr et al., 1988).

At the beginning of the 1920's, total ozone column have been measured by the Dobson sun spectroradiometer (Dobson, 1931; Dobson, 1968, Basher 1982, Komhyr et al. 2002) and were installed worldwide to a global network. The Dobson instruments were operated manually and required substantial manpower and maintenance and were difficult to operate at remote sites. In order to monitor the aforementioned depletion of TCO and its impact on climatic change and the UV exposure on the ground, the Brewer spectrometer (Kerr et al., 1981, Kerr et al. 1985) was introduced in the 1980s as an automatic device measuring

direct solar UV radiation and global UV irradiance with state-of-the-art technology using gratings instead of prisms. Contrary to the Dobson instruments, the Brewer also allowed measuring absolute intensities, in contrast to the Dobson instruments, which provide relative measurements of the different UV wavelength bands. Both, the Brewer and the Dobson instruments were considered as the standard instruments for TCO monitoring on the ground by the Word Meteorological Organisation (WMO) in the framework of the Global Atmosphere Watch program (GAW).

The Dobson as well as the Brewer are operationally using four specific wavelengths in the UV band between 300 nm and 340 nm for the retrieval of ozone with the double ratio technique (Kerr et al, 1988) and have to be calibrated for TCO against a regional standard instrument via on-site intercomparisons (Köhler et al. 2002, Redondas et al. 2018). The intercomparison between the Dobson and Brewer revealed substantial differences. Systematic biases and seasonal dependency were observed when comparing both instruments (Kerr at al., 1988, Köhler et al., 2018, Stählin et al., 2018, Vanicek et al., 2012, Redondas

et al., 2014, Gröbner et al., 2021), displaying fundamental uncertainties in estimating TCO with the Dobson or the Brewer. Recently, Gröbner et al. 2021 showed that these systematic biases between Brewer and Dobson can be reduced by using the cross-section from the University of Bremen (IUP, Serdyuchenko et al, 2014) calculated for the appropriate ozone temperature. In recent years, TCO was also retrieved from the Earth's ground accounting for the spectrum between 300 nm and 340 nm, e.g. with the Phaeton system (Gkertsi et al. 2018), the Pandora system (Herman et al. 2017) or the BTS array spectroradiometer

(Zuber et al. 2018a,b, Zuber et al. 2021).

In this study a scanning double monochromator is used to retrieve TCO from direct solar UV spectral irradiance measurements between 305 nm and 345 nm at a wavelength increment of 0.25 nm. The retrieval of TCO is named here as "full spectrum" retrieval of TCO. Due to the larger amount of spectral information in the atmospheric ozone absorption band compared to the four wavelengths of the Dobson and Brewer, one could expect to measure TCO with lower uncertainties.

The main objective of this study is first to introduce traceable TCO measurements from the world portable reference spectroradiometer for UV radiation QASUME (Gröbner et al. 2005) as a reference for full spectrum TCO retrievals. This includes also a thorough investigation and report of the overall uncertainties for TCO retrieval from full spectrum direct irradiance measurements by the scanning double monochromator QASUME. Traceability means that the instrument is characterized and calibrated in the laboratory with sources that are traceable to primary SI standards with known uncertainties

during the entire calibration chain. Since TCO cannot be measured directly, but only from ground remote sensing, a retrieval model algorithm is needed to derive TCO from the measurements. The uncertainty of the retrieval algorithm is also investigated for the various parameters of the retrieval model. "Traceability" with respect to the retrieval model means a standardized, reproducible and comprehensible process to derive TCO from the solar spectrum. Both, the uncertainty of the spectral measurements and the uncertainties of the retrieval algorithm result in an overall uncertainty of traceable TCO observations.

The sensitivity of the retrieval model is determined for various atmospheric input parameters and then optimized for robust TCO estimations. The presented method can also be considered as a reference algorithm for other instruments measuring UV spectra from calibrated direct sun irradiance as e.g. array spectroradiometers (Zuber et al. 2021).

Finally, the results of the traceable TCO measurements from QASUME are compared with long-term observations from a double monochromator Brewer and a Dobson spectroradiometer at the World Radiation Center (PMOD/WRC), in Davos,

Switzerland, between 20 September 2018 and 15 November 2020. This long-term intercomparison allows investigating the robustness of the traceable TCO measurements from QASUME and detecting potential biases to operational TCO observations from Brewer or Dobson spectroradiometers.



## 2 Instrument and Retrieval Algorithm

### 2.1 QASUME portable UV reference instrument


The transportable reference spectroradiometer QASUME was built for quality assurance of spectral ultraviolet measurements in Europe, as described in Gröbner et al. (2005). Since 2001, the portable device operated at more than 33 stations worldwide to monitor the quality of global UV measurements from other instruments such as e.g. the Brewer spectroradiometer. A thorough revised uncertainty budget for the QASUME instrument was calculated and presented in Hülsen et al. (2016). The

uncertainty budget will be used for the estimation of the overall uncertainty budget of TCO in chapter 3.1.

The QASUME instrument used for traceable TCO measurements presented here, is a scanning double monochromator consisting of a commercially available Bentham DM-150 with a focal length of 150mm and a grating with 2400 lines/mm. The spectral range covers the wavelength region between 250 nm and 550 nm. The slit function is almost triangular with a full

width half maximum resolution of 0.78 nm. The entrance optics is connected by an optical fiber to the instrument and is mounted at the end of a collimating tube of 1000 mm length, providing a full field of view of 2°. The collimating tube with the global entrance optics is mounted on a sun tracker following the sun during daytime to collect direct sun irradiance only. The system is embedded in a temperature stabilized box, while the entrance optics is heated to a temperature above 28°C to exclude temperature effects of the entrance optics. The temperature stabilization ensures outdoor measurements during winter

– and summertime at the measurement platform at PMOD/WRC, Davos, Switzerland at 1560 m a.s.l. (coordinates: 46.81 N, 9.83 E).

For this study, direct solar UV measurements were collected with QASUME between 20 September 2018 and 15 November 2020 during all seasons, but with some timeout periods, when the system was in the laboratory or participated at operational measurement campaigns. Spectral measurements of QASUME are traceable to the primary 1000 W tungsten halogen standard

lamp of the Physikalisch Technische Bundesanstalt (PTB) (Gröbner and Sperfeld, 2005), and the stability of QASUME was monitored with traceable 250W halogen lamps on a regular schedule on the measurement platform.

The spectra were recorded regularly on an interval of 30 minutes and occasionally on an interval of 20 min. Since the instrument is a scanning spectroradiometer, the time interval between each wavelength increment of 0.25 nm is less than or equal 1.5 seconds, therefore the measurement of the spectrum between 305 nm and 345 nm requires maximum 4.5 minutes.

More than 3200 spectra, valid for TCO retrieval, were recorded from morning to evening at different solar zenith angles (SZA) ranging between 23° and 76° (air mass between 1.1 and 4.0) during 20 September 2018 and 15 November 2020. The measured spectra were used for the TCO retrieval algorithm as describes in the following section.

### 2.2 Total ozone column retrieval algorithm


The post processing of the calibrated UV spectra from QASUME was performed off-line in two steps: a) wavelength shift correction and homogenisation of the spectra with the MatSHIC software developed at PMOD/WRC, b) retrieving TCO with a least square fit (LSF) minimization algorithm according to Huber et al. (1995).

The following presented post-processing chain displays a "traceable" algorithm routine, which ensures consistent and

comparable TCO measurements derived from spectral solar measurements.

### a) MatSHIC algorithm

The MatSHIC algorithm is a Matlab version based on the SHICrivm algorithm developed by Slaper et al. (1995). The algorithm convolves a high resolution reference solar spectrum (ETS) with the slit function of QASUME and determines the spectral

shift for each wavelength of the measured spectrum until the best agreement to the convolved ETS is found. The so detected


wavelength shift is then applied to each wavelength of the measured spectrum to provide consistency to the reference solar spectrum. In a second step, the algorithm adjusts the high resolution ETS to the wavelength corrected measurements. Since the ETS is a high-resolution spectrum with 0.01 nm wavelength steps and around 0.05 nm full width half maximum slit function, the resulting spectrum can be homogenized by convolution with a triangular slit function to a selectable wavelength

increment and slit function, equal or larger than the wavelength-increment and slit function of the ETS. For this study, the resulting spectra were homogenized to 0.01 nm wavelength-increment and convoluted to 0.5 nm full width half maximum with a triangular slit function. These selected specifications have been shown to be the best homogenization settings for the TCO retrieval in terms of overall uncertainty.

*b) Total ozone column retrieval algorithm from full UV spectrum*

For the full spectrum TCO retrieval the LSF algorithm presented by Huber *et al.* (1995) and applied in Vaskuri *et al.* (2018) and further described in Zuber et al. (2021) is refined and optimized here for traceable TCO measurements with QASUME. The LSF algorithm is using a full spectral fit in the main UV ozone absorption wavelength range between 305 nm to 345 nm and applies an atmospheric model based on the Beer-Lambert law:


$$I_\lambda = I_0 \, exp[-\tau_\lambda m] \qquad \text{Eq.1}$$

$I_\lambda$ denotes the measured spectral irradiance from QASUME and homogenized by MatSHIC at the wavelength $\lambda$. $I_0$ indicates the ETS at the top of the atmosphere. For the standard retrieval presented here, the new composite hybrid solar reference

spectrum TSIS (Coddington et al., 2021) is used for $I_0$. Finally, $m$ is the airmass from Earth's surface to the top of the atmosphere.

The atmospheric model accounts for the effect of ozone absorption for each wavelength and therefore the attenuation of the ETS by the atmosphere including aerosols, Rayleigh scattering and SO2. The resulting attenuated spectrum is then compared with the measured spectrum on earth surface. The LSF algorithm minimizes the residuals between the modelled and measured

solar spectrum and returns the corresponding model parameters. Figure 1 shows the residuals for each wavelength for an exemplarily day of 27[th] of June 2020. The figure shows that the residuals are spectrally flat with a high spectral variation between 300 nm and 305 nm and a slight increase at wavelength larger than 345 nm. The spectral variation of up to 2% seen in Figure 1 are residuals from the uncertainties in the convolution of the high resolution ETS with the measured QASUME slit function during the MatSHC procedure.

An important parameter of the model in Eq.1 is the airmass $m$, denoting the path length of radiation through the atmosphere. The airmass $m$, depends on the geographical location and time of the day and thus on the solar zenith angle during a day and over the seasons. The airmass is calculated based on the geometry between the Earth, atmosphere and the sun for each time stamp and corresponding wavelength of the measured spectrum individually. The absorption through the atmosphere is summarized by the term $\tau(T)_\lambda m$ in Eq. 2. The term $\tau(T,p)_\lambda m$ indicates the attenuation of direct irradiance by ozone, aerosols,

sulfurdioxide (SO2) and Rayleigh scattering during its path through the standard atmosphere. The airmass for the ozone ($m_\lambda^{O3}$), aerosol ($m_\lambda^{AOD}$) and Rayleigh ($m_\lambda^{R}$) and SO2 ($m_\lambda^{SO2}$) is calculated from the standard US atmosphere profile for mid-latitudes *afglus* (NOAA, 1976) and Eq. 1 can be written in more detail as Eq. 2.

$$\tau_\lambda(T,p) \cdot m_\lambda = \tau_\lambda^{O3}(T) \cdot m_\lambda^{O3} + \tau_\lambda^{AOD} \cdot m_\lambda^{AOD} + \tau_\lambda^{R}(p) \cdot m_\lambda^{R} + \tau_\lambda^{SO2} \cdot m_\lambda^{SO2} \qquad \text{Eq.2}$$


For the ozone attenuation term $\tau_\lambda^{O3}(T) \cdot m_\lambda$ the IUP ozone absorption cross section (Serduychenko et al. 2014) for different effective ozone temperatures ($T$) parametrized with a quadratic polynomial fit is applied for the standard retrieval algorithm. This cross section has been selected by the WMO as the future new reference cross sections for the Brewer and Dobson (M.





Tully, personal communication in Gröbner et al. (2021)) and the best consistency between Brewer and Dobson is found for

the IUP cross section (Redondas et al. 2014, Gröbner et al. (2021). However, the effect of other available cross sections will be analysed and discussed in section 3 addressing the overall uncertainty budget. The dependency of ozone absorption on effective ozone temperature implies, that the effective ozone temperature is required for the TCO retrieval algorithm. In analogy to Gröbner et al. (2021), the effective ozone temperature measurements from balloon sounding at the nearest sounding station in Payerne, Switzerland is taken as input for the retrieval algorithm.

The term $\tau_\lambda^{AOD} \cdot m_\lambda$ in Eq. 2 denotes the attenuation by aerosol optical depths. The wavelength dependence of the aerosol optical depth ($AOD$) for $\tau_\lambda^{AOD} \cdot m_\lambda$ is defined with a linear parametrization normalized to 340 nm as follows

$$AOD = \; \alpha \; + \; \beta \cdot (\lambda - 340\text{nm}) \qquad \text{Eq. 3}$$

where α and β are the other two free parameters besides TCO for the LSF retrieval. The linear parametrization of the AOD is also implemented in the double ratio technique for the Brewer retrieval (Kerr et al. 1985).

The third term in Eq. 2 $\tau_\lambda^R(p) \cdot m_\lambda$ accounts for the Rayleigh scattering in the atmosphere. The scattering by Rayleigh is parametrized according to Bodhaine et al. (1999). This parametrization requires the air-pressure $p$ on the ground to determine the amount of Rayleigh scattering in the atmosphere. Finally, the attenuation by SO$_2$ should also be considered in the overall

attenuation equation (Eq.2). $\tau_\lambda^{SO2} \cdot m_\lambda$ is parametrized with the parametrization used for the Brewer by Kerr (1985).

Accounting for all four terms of attenuation in the Beer-Lambert atmospheric model, the LSF approach derives the best fit to determine TCO within the free parameters.

In summary, the standard full spectrum TCO retrieval with QASUME consists of the following settings:


- MatSHIC spectrum homogenization:     0.5 nm FWHM, 0.01 nm wavelength increment
- Wavelength range:     305 nm – 345 nm
- Ozone absorption cross section:     IUP (Serduyschenko et al. 2014)
- Extraterrestrial Spectrum:     TSIS (Coddington et al. 2021)
- Aerosol optical depth (AOD):     Linear spectral function
- Rayleigh scattering     Bodhaine (Bodhaine et al. 1999)
- SO$_2$     Kerr (Kerr et al. 1985)
- Effective ozone temperature:     Input parameter from balloon soundings (Gröbner et al. 2021)
- Atmospheric model:     Beer-Lambert law (Eq.1), with US standard atmosphere (NOAA, 1976)


### 3 Uncertainty Budget

#### 3.1 Measurement Uncertainty

The measurement uncertainty of QASUME is well reported in Hülsen et al. (2016) for global UV measurements and recalculated for direct solar irradiance in Gröbner et al. (2017). The different contributions for the direct UV measurement

uncertainty are separately listed in the publication resulting in an overall uncertainty of 0.91% of QASUME direct measurements (Gröbner et al. (2017), Vaskuri et al. 2018). Vaskuri et al. (2018) stated an uncertainty of 0.38% when considering this random noise of 0.91% from the measurements. The uncertainty assessment in Gröbner et al. (2017) are not reporting any spectral correlation of the uncertainty, which may occur due to the calibration process with standard lamps. The effects of spectral correlations on full spectrum retrievals are investigated and discussed in Vaskuri et al. (2018) applying





sinusoidal spectral correlations of different degrees. Depending on the degree of spectral correlation, the uncertainty for TCO from full spectrum originating from the measurements can result in uncertainties of TCO between 0.72% (full correlation) 0.42% (unfavorable correlation) and 0.38% (no correlation) (Vaskuri et al. 2018). Assuming no or unfavorable spectral correlation we obtain an uncertainty of TCO originating from spectral measurements from QASUME of 0.42% (Table 1).

**3.2 Uncertainty from ozone absorption cross section**

In order to estimate the uncertainty derived from the ozone absorption cross sections, we compare TCO retrieved with the standard settings but with four other available cross sections. For comparability, the same cross sections with the same quadratic or linear parametrizations of the effective ozone temperature as in Gröbner et al. (2021) and available from the IGACO webpage (http://igaco-o3.fmi.fi/ACSO/cross_sections.html) are used and summarised as follows:

IGQ    The Bass&Paur (Bass and Paur, 1985) ozone absorption cross sections from the IGACO web-page, with a quadratic parametrisation of cross section temperature (file bp.par).

DBM    Daumont et al. (1993), Brion et al. (1993), and Malicet et al. (1995) published a high resolution dataset of at five temperatures between 218 K and 295 K. As in Gröbner et al. (2021) a linear parametrisation of the ozone temperature 225    is applied, due to the lack of measurements at temperatures below 218 K.

IUP_A   This absorption cross sections measured by the University of Bremen, IUP in 2017 (M. Weber, personal communication in Gröbner et al, 2021) during the project European Metrology Research Program ATMOZ (Traceability of atmospheric total column ozone) led by PMOD/WRC. Here and in Gröbner et al. (2021) the quadratic 230    polynomial temperature approximation is used.

ACS    Birk et al. (2018) measured a new cross section in the frame of the ESA project SEOM-IAS between 243 nm and 346 nm. The temperature range is at 193 K, 213 K, 180 233 K, 253 K, 273 K, and 293 K. As for IUP a quadratic polynomial temperature dependence fit is applied to parametrise the temperature dependency.


Figure 2 presents the relative differences between TCO retrievals with the aforementioned specific cross sections and the resulting TCO derived with the IUP cross section. The figure shows the relative difference of over 3200 TCO data recorded between 20 September 2018 and 15 November 2020. The subfigures display the mean of the differences, indicated as offset, the standard deviation of all data points and the seasonal variation in terms of a sinusoidal fit. TCO from the different cross 240 sections deviate in average between -0.43% and +0.31% for IGQ, DBM, IUP_A cross sections. The results of the ACS cross section demonstrate that this new cross section denotes an outlier compared the other comparison. This outlier is also remarked in Gröbner et al. (2021). Based in the previous results and the results here, we assume that this new cross section still includes some inconsistencies in the considered wavelength range. Therefore, ACS is excluded here for the uncertainty analysis origination from cross section.

To account for the variability of each individual measurement (indicated by the amplitude) and the offset, all relative differences from the three cross sections are merged and the mean and standard deviation is calculated resulting in an offset of -0.15% and a standard deviation of 0.38%. We therefore state and note in Table 1 an uncertainty of TCO retrieval from the cross sections of 0.38% (k=1).





### 3.3 Uncertainty from extraterrestrial spectrum

In analogy to the uncertainty assessment of the cross sections we derive the uncertainty of TCO resulting from ETS by including a second ETS. As in the previous section, the other standard parameters are kept constant for the retrieval. The second ETS is called QASUMEFTS which is derived from QASUME measurements and high-resolution Fourier Transform Spectrometer (FTS) at the Izana Observatory, Teneriffe, Spain during September 2016 and fully described in Gröbner et al (2017) and compared in Coddington et al. (2021) with the TSIS solar spectrum. The QASUMEFTS exhibits an overall uncertainty of around 1% (k=1) for the wavelength range between 310 nm and 350 nm and gradually about 2.0% (k=1) between 300 nm and 310 nm as reported in Gröbner et al. (2017) and is therefore comparable with the uncertainty from TSIS with an uncertainty of 1.3% at wavelengths shorter than 400 nm (Coddington et al., 2021).

Figure 3 presents the comparison of TCO retrievals from TSIS versus QASUMEFTS (offset= 0.68%, standard deviation= -0.24%). Since only two ETS are compared, the offset of 0.68% displays the size of a rectangular distribution. The uncertainty for k=1 is therefore given by $0.68/\sqrt{3}$ % = 0.38% (Table 1). The resulting bias of 0.68% between TSIS and QASUMEFTS may be caused by spectral correlations of the QASUME measurements and the solar spectrum TSIS which are not explicitly known. This bias is in the order of 0.72%, as reported in Vaskuri et al. (2018), when a full correlation of measurement uncertainties between the spectral measurement and the extraterrestrial spectrum is assumed. The effect of the measurement uncertainty on TCO retrieval is about 0.42% for "no correlations" (Table 1). Potential spectral correlations of the measurement uncertainty are therefore reflected here by the comparison of two different ETS.

### 3.4 Uncertainty from effective ozone temperature

Figure 4 shows the temperature dependency of the QASUME TCO retrieval algorithm normalized to 228 K which is usually taken as a climatological value for Brewer retrieval. The calculated temperature dependency shows that the sensitivity of TCO is about 0.1 %/K for IGQ, IUP_A and IUP cross section and about 0.08%/K for ASC and only 0.04K/% for DBM cross section. These values are comparable with the temperature sensitivity of the Dobson (Gröbner et al. 2021, Redondas et al. 2014), while the Brewer show almost no temperature dependency (e.g. for the IUP cross section).

The effective ozone temperature for Davos is derived from balloon soundings in Payerne Switzerland, 220 km distant from Davos on a two- or three days schedule. Gröbner et al. (2021) stated a seasonal variability of the effective temperature of amplitude of 11.4 K and a mean value of 225.2 K between 2016 and 2020. This means that the seasonal standard deviation is about 5K, indicating a seasonal temperature uncertainty, which is correlated with time during the seasons. To reduce the uncertainty of TCO from effective ozone temperature, however, we included the temperature in the retrieval instead of a climatological value. Gröbner et al. 2021 also compared the effective ozone temperature from balloon soundings with ECMWF reanalysis data from (http://www.temis.nl/climate/efftemp/o...) with daily data and revealed a standard deviation of 2.5 K for the period between 2016 to 2020. The differences between the two datasets are not correlated within the seasons and we therefore pragmatically consider the uncertainty of observing the temperature as 2.5 K, when using either balloon soundings or ECMWF reanalysis data. Considering this sensitivity on effective temperature of 0.1K/% for QASUME TCO measurements and the estimated uncertainty of measuring the effective ozone temperature, a general uncertainty for TCO of 0.25% (k=1) can be stated from effective ozone temperature and included in Table 1.

### 3.5 Air mass

The standard retrieval algorithm includes the US standard atmosphere *afglus* with defined ozone profile. However, balloon soundings from Payerne, Switzerland, show that the effective ozone height can change to about 3.6 km within the seasons (Gröbner et al., 2021). For the specific location in Davos and during the time of the period analyzed here, a maximum deviation





of the airmass of 0.3% of TCO due to the change of the effective ozone height is assumed. Following Eq.1 the airmass affects the uncertainty of TCO linearly. Since 0.3% displays the maximum deviation an uncertainty from variable airmass can be estimated as about of $0.3\%/2\sqrt{3}=0.086\%$ (k=1) (see Table 1).

### 3.6 Uncertainty from ground Pressure

The sensitivity of air-pressure of retrieved TCO is about 0.2% for pressure changes of 100 hPa. In average, the air-pressure in Davos was around 840 hPa and varied only between +/-7 hPa during the period of comparison, which results in a variation of TCO of 0.014%. Due to this small sensitivity of TCO on air pressure and to simplify the standard algorithm, a constant air pressure of 840 hPa was used as input parameter for the algorithm. The resulting variation of TCO is here considered as an TCO uncertainty of 0.014% from Rayleigh scattering parametrization with variable ground air pressure (Table1). For locations

with higher variations of air pressure, the measured pressure can be taken as the input for the algorithm.

### 3.7 Uncertainty from Least Square Fit Algorithm (Computational)

As a criterion for valid TCO retrieval, residuals of the Jacobian matric from the in-built Matlab function "lsqnonlin" are calculated indicating the 95% confidence interval of the retrieved TCO. If the value is less than 0.7 DU (or about 0.23 % at 300 DU) then the retrieval is considered as valid. If the value exceeds 0.7 DU, the measurement could have been disturbed by

moving clouds, overcast sky or other atmospheric effects. The criterion of 0.7 DU indicates that TCO varies less than 0.25% (at 300 DU). Therefore, we define the maximum relative uncertainty of the least square fit retrieval model as 0.25%/2=0.125 %. (k=1, Table 1).

### 3.8 Overall Uncertainty

In section 3.1 to 3.7 the individual sources of uncertainty are calculated and listed in Table1. In order to estimate an overall

uncertainty two methods are chosen:

a) *Arithmetic calculation of overall uncertainty*

Assuming gaussian noise and no correlation of the individual uncertainties as listed in Table 1, the overall uncertainty can be calculated by the square root of the sum of the quadrate from individual uncertainties:

$$\hat{u} = \sqrt[2]{\sum u_i^2} \qquad \text{Eq. 4}$$

Equation 4 results in a calculated overall uncertainty of 0.74% when including all 7 contributions (Table 1).

b) *Monte Carlo Simulation*

Since TCO is derived by an atmospheric model as described in section 2.2 the overall uncertainty of the model and the corresponding input parameters can also be determined by a Monte Carlo simulation (Vaskuri et al., 2018). Monte Carlo

simulation means that all the input parameters of the model, such as measurement, cross sections, effective ozone temperature, solar spectrum, pressure and airmass are varied within their specific uncertainties as listed in Table 1. Each variation displays a realization of possible TCO retrievals. Specifically, the TCO time series between 20 September 2018 and 15 November 2020 is recalculated for each of the 3200 individual measurements by a random variation of the different retrieval model parameters. The entire time series is recalculated 10 times to obtain a total number of randomly varied realizations of more than 32'000.

Figure 5a presents the standard deviation over 10 realizations for a single measurement (blue points). The black line indicates the mean over all 3200 measurements of 0.62 +/- 0.16% (black dashed lines). Since the uncertainty may be dependent on solar zenith angle and the corresponding air mass change, the uncertainty of each measurement is displayed as a function or air mass (Figure 5b). The cubic fit highlights that the uncertainty is about 0.16% larger for lower air masses than for airmasses larger than 4. The analysis of 10 realizations may not be sufficient to assess the maximum overall uncertainty. Therefore, the standard

deviation over the entire period with 32'000 variations is computed. Figure 6 presents the frequency distribution of the





differences between TCO from the standard algorithm and TCO from varied input parameters. The mean of the distribution is around 0.34% with a standard deviation of 0.80%, which is nearly gaussian distributed (Figure 6, red line). The bias of 0.31% originates from the differences between the two selected ETC's (Figure 3), which differs of 0.39 % (Table 1). Other biases of individual uncertainties contribute neglectable to the observed bias.

Combining the two statistical analysis of the Monte Carlo simulation (Figures 5 and 6), we achieve an average uncertainty of 0.71%. This simulated uncertainty is close to the uncorrelated uncertainty of 0.74 % according to Eq. 4. The similarity of the uncorrelated and the Monte Carlo uncertainty indicates that the individual uncertainties are weakly correlated. However, they are slightly depending on the airmass. Therefore, Eq. 4 can be applied to assess the overall uncertainty budget by combining the individual uncertainties. The reported relative expanded uncertainty of measurement is stated as the standard relative

uncertainty of measurement multiplied by the coverage factor k = 2, which for a normal distribution corresponds to a coverage probability of approximately 95%. From our calculations, the upper limit of the overall uncertainty from traceable TCO retrievals from QASUME is given by the maximum of the Monte Carlo simulation of 1.6% (k=2). This maximum uncertainty can be considered as a benchmark uncertainty of TCO estimates with a traceable calibrated and characterized reference instrument.


## 4 Discussion

*Comparison with Dobson and Brewer*: Gröbner et al. (2021) presented a long-term comparison between automated Dobson (Stübi et al. 2020) and Brewer from the Lichtklimatologische Observatorium Arosa (LKO) and PMOD/WRC Davos TCO time

series. The comparison accounted for the different cross sections as used here, the physical characteristics of the instruments (e.g. measured slit functions) and the effective ozone temperature. The comparison revealed that the best agreement can be found when using the IUP ozone absorption cross section (Serdyuchenko et al., 2014) as it is defined here for the standard retrieval. The comparison of TCO from Brewer and Dobson with other ozone absorption cross sections, however, revealed variation of up to 5% between these two instruments. Since only two instruments were compared, it could not be explained,

which of both instruments is more sensitive on the selection of the absorption cross section.

With QASUME we have introduced a third independent type of instrument for measuring TCO. In analogy to Gröbner et al. (2021), Figure 7 presents the comparisons of the Brewer 156 double monochromator and Dobson 101 with QASUME TCO for five different cross sections. The results show that averaged TCO ranges between -4.06% (DBM) and 0.98% (IGQ), when comparing QASUME with the Brewer 156. This large inconsistency of about 5% between the use of different ozone absorption

cross sections was also reported by Gröbner et al. 2021 analysing the comparison between Brewer and Dobson only.

The comparison between Dobson and QASUME ranges between -1.01% (IUP) to - 0.72 (DBM), which is significantly smaller than the variation of the Brewer. The variability of 5.04% for the Brewer-QASUME comparison and the much lower variability of 0.29% for the Dobson-QASUME comparison proofs that the Brewer is more sensitive on the selection of the ozone absorption cross section than the Dobson or QASUME. Without the comparison with the alternative TCO retrieval with

QASUME it would remain unclear if the Dobson or the Brewer is more sensitive to the selected absorption cross section.

As concluded by Gröbner et al. (2021), the IUP cross section shows the best consistency between Brewer and Dobson of less than 0.21%. However, Figure 7 shows that QASUME over-estimates TCO by about 1% (Dobson) or 1.2% (Brewer).

The minimal least square fit TCO retrieval algorithm of QASUME "full spectrum" is a distinct different retrieval approach compared to the double ratio technique for Brewer and Dobson. This independent instrument and retrieval confirm the ground

based TCO retrieval from the established instruments within 1%. Additionally, Figure 8 presents the dependency of the differences of TCO on the slant path column, which is defined as the multiplication of air mass and TCO. The graph reveals that compared to the Brewer 156 double monochromator, QASUME TCO shows almost no dependency on the slant path. Since QASUME consists of a double monochromator, the straylight suppression is sufficient to measure TCO without biases




due to straylight. In contrast and reported in (Komhyr et al., 1986), the Dobson suffers from straylight effects at slant path
larger than 900 DU, compared to double monochromator instruments. TCO from QASUME combines the advantage of
insensitivity on ozone absorption cross section as for the Dobson, with negligible straylight effects as for the Brewer double
monochromator. However, the bias of 1% of QASUME compared to Brewer and Dobson and the variation of the individual
measurements (Figure 6) between QASUME and Brewer or Dobson indicates a disadvantage towards a potential independent
short-term field calibration of Brewer and Dobson with traceable TCO observations. The sources of the short-term deviations
(e.g. weather conditions, cirrus clouds) may be investigated in more detail. So far, QASUME serves at least as an independent
instrument to measure TCO traceably within 1% and to monitor the relative long-term stability of Brewer and Dobson within
an uncertainty of less than 0.8%.

## 5 Conclusion

We have introduced traceable TCO ground based retrievals from direct solar UV spectral measurements in the wavelength
range between 305 nm and 345 nm. The instrument is calibrated in the laboratory based on a traceability chain from SI
standards (lamps) and the retrieval algorithm is standardized and optimized for low overall uncertainties. The overall standard
uncertainty of 0.8% is quantified by the uncertainties from the radiometric quantities, the instrument characteristics and the
various uncertainties of the retrieval model. The expanded relative uncertainty of 1.6% indicates the standard overall
uncertainty multiplied by the coverage factor k=2 of the gaussian distribution and corresponds to a coverage probability of
95%. The Monte Carlo simulation of the overall uncertainties revealed that the sources of uncertainties are weakly correlated
and the overall individual uncertainty can be calculated by the square root of the sum of all individual uncertainties.
The calibration and stability of the instrument can be monitored at any time in the laboratory and the retrieved TCO is
independent of comparisons with reference instruments, while the Dobson and Brewer require periodic re-calibrations by a
field comparison with a reference instrument or Langley calibration at locations with stable atmospheric conditions.
The Arosa/Davos time series consist of a triad of Dobson and a triad of Brewer (Stählin et al. 2018). This triad of two instrument
types is now complemented with a laboratory calibrated and characterized instrument providing traceable TCO readings from
a distinct different TCO retrieval algorithm. When using the IUP ozone absorption cross section, QASUME agrees with Brewer
and Dobson with an offset of 1% in a long-term comparison in Davos. Further studies may investigate the performance of
QASUME at other locations worldwide and analyse the restrictions of short-term comparisons due to weather conditions such
as cirrus clouds or large aerosol optical depths.

*Competing interests.* The authors declare that they have no conflict of interest.

*Acknowledgement:* This research has been supported by the ESA project QA4EO, grant no. QA4EO/SER/SUB/09) and by
GAW-CH MeteoSwiss (project INFO3RS, grant no. 123001926).

*Author contributions:* LE and JG developed the QASUME retrieval algorithm, analysed the data and have written the
manuscript. GH operated QASUME at PMOD/WRC and contributed to the manuscript. HS and RS were responsible for
Brewer and Dobson measurements and contributed to the manuscript.

*Data availability:* The data is available from the main author (LE) on request.

*Code availability:* The MatSHIC and TCO retrieval algorithm is available from the main author (LE) on request.






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





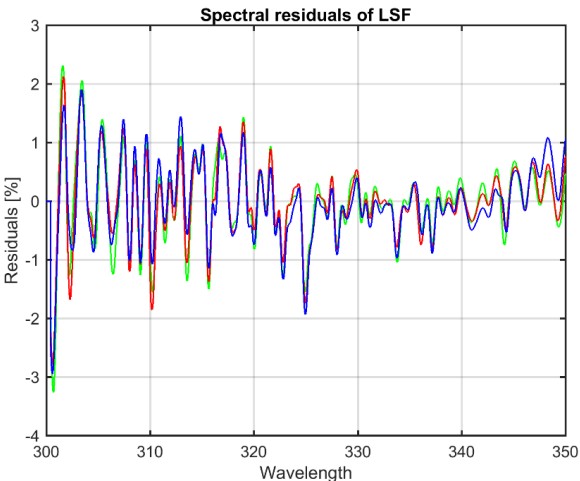

Figure 1: Spectral residuals of the measured and the modelled spectra of the best fit from the LSF algorithm. The exemplary figure shows three spectra at local noon and retrieved TCO of 303 DU. The average of the residuals of the irradiance over all wavelengths are less than 0.5%.

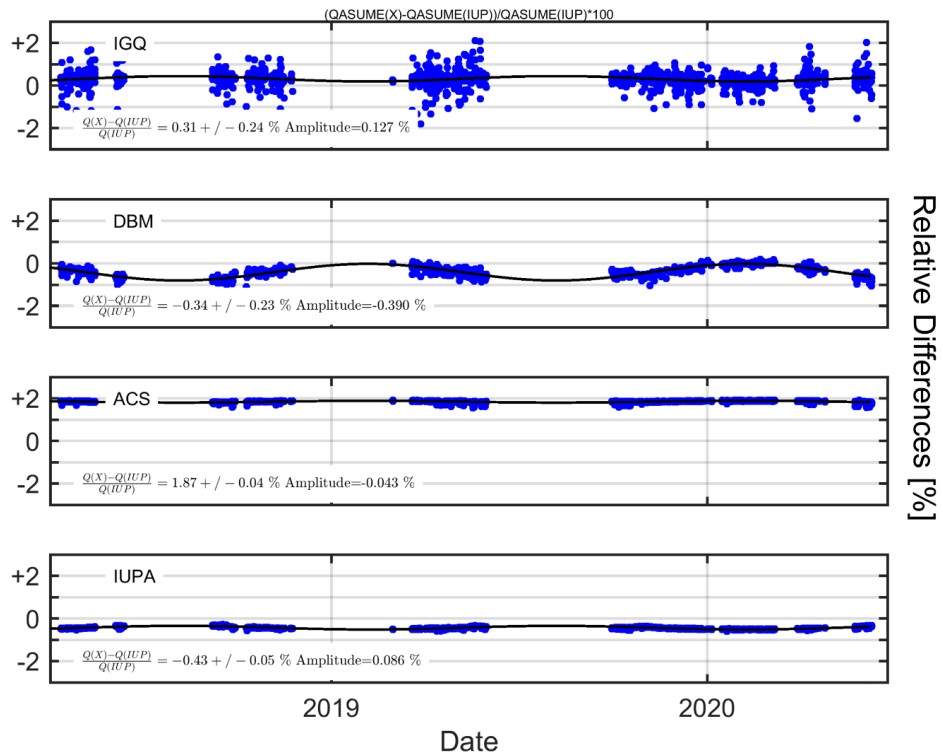

Figure 2: Differences (in %) of systematic offset, point to point variability (+/-) and seasonal variation (amplitude) of TCO retrieval of different cross sections compared with the IUP standard cross section.





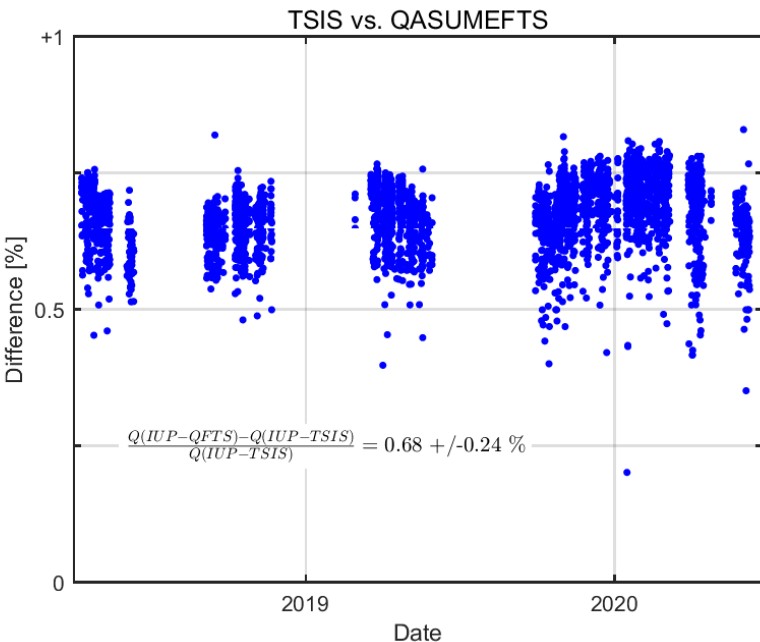


*Figure 3: Differences between TSIS extraterrestrial spectrum and QASUMEFTS with IUP cross section displaying the uncertainty originating from the selection of the solar spectrum.*

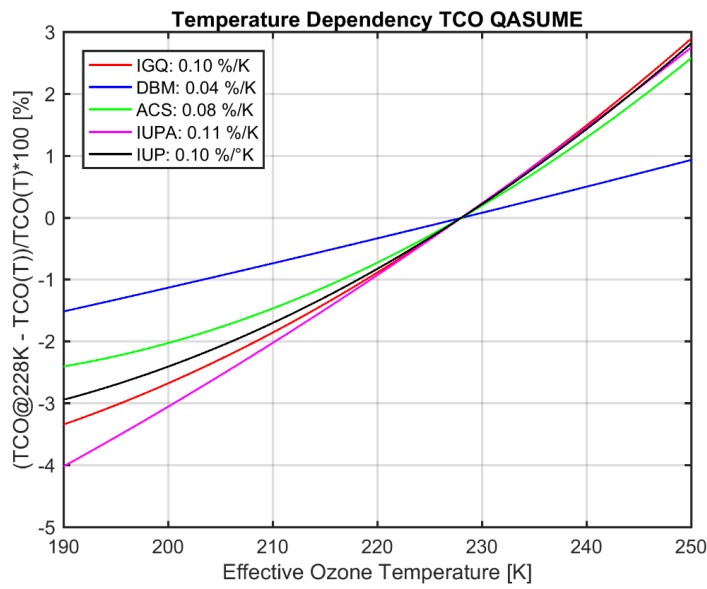


*Figure 4: Dependency on effective ozone temperature of five different cross sections*





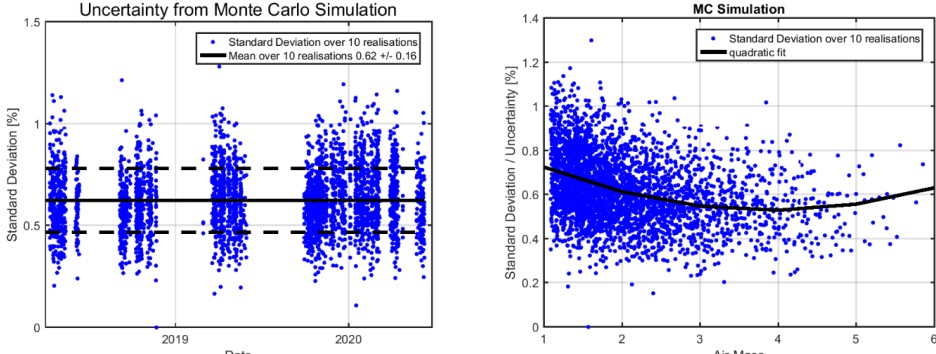

Figure 5: Results of the 32'000 realisations of TCO from Monte Carlo simulation (MC). The left panel shows the standard eviation of 10 realizations for each measurement points and the right panel indicates the dependency on airmass


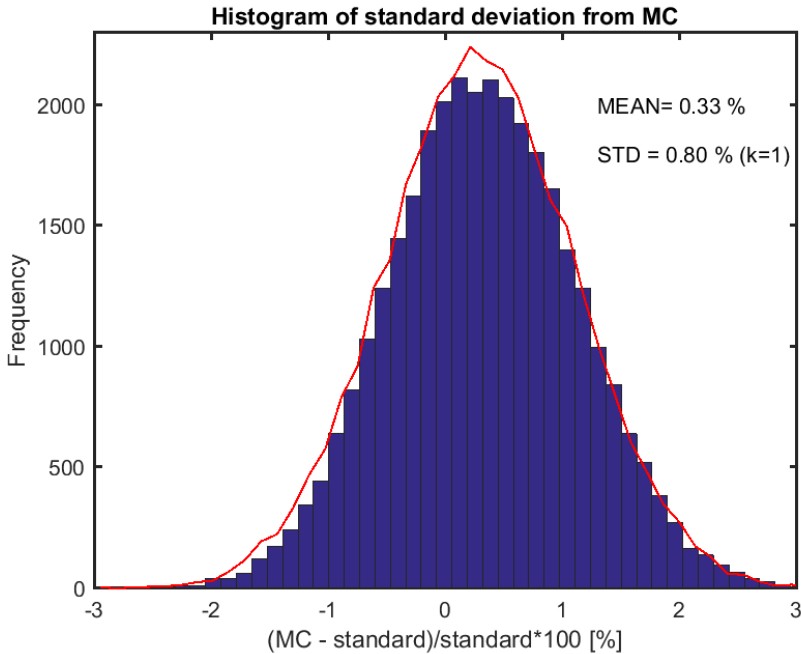

Figure 6: Histogram of all realizations from the Monte Carl Simulation indicating an overall uncertainty of 0.8% (standard deviation of differences) of traceable TCO measurements with QASUME. The red line indicates a randomly generated gaussian distribution.





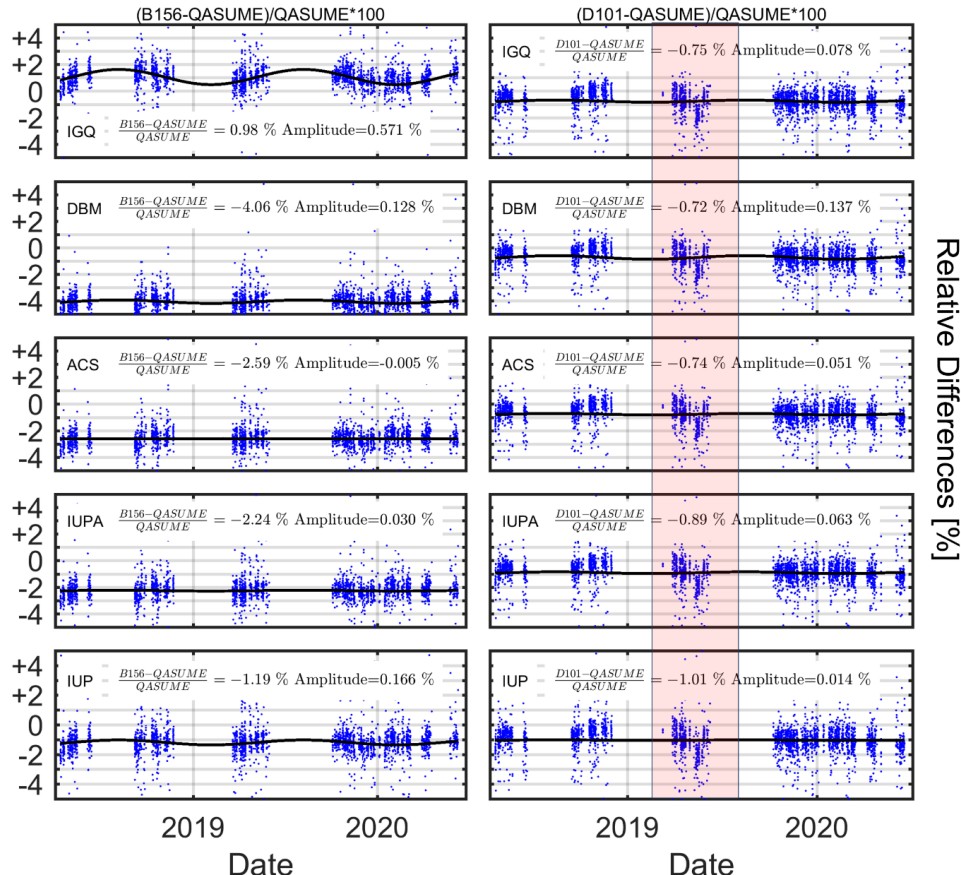

*Figure 7: Comparison of TCO from Brewer 156 (left panels) and Dobson 101 (right panel) with TCO from traceable QASUME TCO retrieval for five*

*different cross sections. The red bar indicates that the Dobson-QASUME comparison is consistent for all tested ozone absorption cross section.*

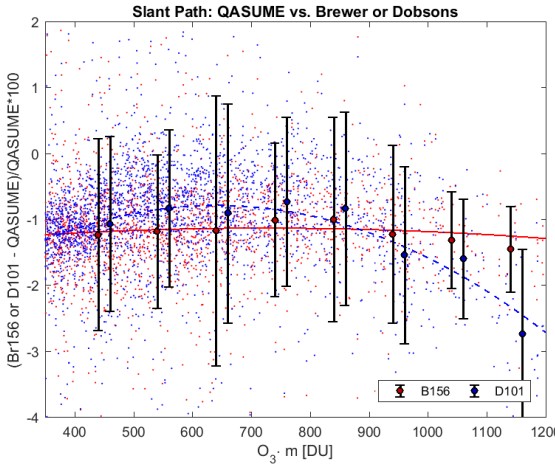

*Figure 8: Differences of Brewer 156 and Dobson 101 compared with QASUME TCO depending on the ozone slant path. The comparison with Brewer*

*shows neglectable straylight effects.*





| Parameter | Type | Input Uncertainty | Standard Uncertainty of TCO (k=1) |
|---|---|---|---|
| Measurement | Gaussian | 0.91% | $u_1$ = 0.42 % |
| Ozone absorption cross section | Rectangular | 4 cross sections | $u_2$ = 0.38 % |
| Effective Ozone Temperature | Gaussian | 0.1%/K for +/-2.5°C | $u_3$ = 0.25% |
| Computational | Rectangular | <0.25% (k=2) | $u_4$ = < 0.125% |
| ETS | Rectangular | 2 ETS's = 0.68% | $u_5$ = 0.39% |
| Pressure | Gaussian | 0.002%/hPa for +/- 7 hPa | $u_6$ = 0.014% |
| Ozone air mass | Rectangular | 0.3% (k=2) | $u_7$ = < 0.085% |
| Total (calculated) | $\hat{u} = \sqrt[2]{\sum u_i^2}$ | | $\hat{u}$ = 0.74% |
| Total (Monte Carlo) | 32'000 runs | | $\hat{u}_{MC}$ = 0.62 +/- 0.16 to max. 0.80 % |
| Overall Uncertainty | Gaussian | | $\hat{u}_{tot}$ < 0.8 % (k=1), 1.6% (k=2) |

*Table 1: Sources of the overall uncertainty and its effect on the resulting overall TCO uncertainty*
