# Peer review of "Traceable total ozone column retrievals from direct solar spectral irradiance measurements in the ultraviolet"

_Atmospheric Measurement Techniques, 2021_

## Referee Comment (RC1)

**General Comments:**

The paper by Egli et al. presented research on deriving traceable total column ozone with the QASUME instrument. Traceability, or more specifically, a detailed uncertainty budget, is in great need for ground-based ozone observations. The methodology and analysis applied are comprehensive and solid. The instrument shows a good potential to be an independent/third-party reference to Brewer and Dobson ozone observations. However, some of the results still need more investigation or clarification. For example, the -1% offset between QASUME and Brewer or Dobson should be investigated (or, at least better described with more details), as the author claimed a 0.8% standard uncertainty. Overall, the paper is a well-written one. I recommend publishing it on AMT after addressing the following comments.

**Specific Comments:**

L63-64: please consider changing "the four wavelengths of … " to "the standard four wavelengths of … ". Please note that Kerr (Kerr, 2002) developed a scanning TCO retrieval method for the Brewer, and published the work back in 2002.

L159: If the term $\tau$ is a function of T and p, please make it consistent. In this line, the term was referenced as $\tau(T)$ and $\tau(T,p)$. Please clarify.

Eq. 3: Is this should be $\tau^{AOD}$? (i.e., not "AOD = … ")

L212-213: Some justification of this assumption is needed.

L227: What is the major difference between these IUP_A and IUP cross-sections? Some description is needed.

L233: This could be a typo, i.e., "180 233 K". Please double check.

L248: The meaning of "(k = 1)" is not described (same request to the "k=2" in the later part of the paper).

L269-274: It might not be very straightforward for the reader to understand why IUP was selected (by WMO), if DBM shows less temperature dependency. Could you please provide some comments? For example, as I know, Pandora is using DBM.

L277-279: I am a bit confused here. Is the effective T been retrieved or not? If it is retrieved, the results should be presented, and some description is needed.

L 326-328 and Figure 5: The right panel has y label as "Standard Deviation/Uncertainty". What is this "uncertainty"? Is this the nominal one (i.e., "arithmetic results")? But the lines in the paper described it simply as "uncertainty", not "Standard Deviation/Uncertainty". Please clarify.

L358-359: I think this "averaged TCO" should be "averaged relative difference (of TCO)".

L362-364: The results here are very interesting and important. I do not want to be hypercritical. However, this information/claim might be very misleading too. First, lower variability for the Dobson-QASUME (D/Q) pair only can prove they (these two instruments) have a similar response (to the set of ozone cross-sections studied here). Second, the results depend on which cross-section is used. If we only select IGQ and IUP, the differences are very small (I doubt if the difference is statistically significant). Is this sensitivity due to instrument or due to cross-sections, or both? Anyway, the point is there is only one true ozone value. For any instrument, being sensitive to a "non-ideal" ozone cross-section might not be a bad thing. I would suggest rephrasing this part as "Brewer is more sensitive to some of the ozone absorption cross-sections (e.g., … ) than … ".

Figure 7. Based on previous publications, Brewer has a remarkably low temperature-dependency (e.g., Kerr 2002), or at least "theoretically" better than Dobson (even with Bass&Paur, or IGQ here). However, the top panel in Figure 7 show that QASUME agrees better with Dobson; it shows a stronger seasonal structure when compared with Brewer. To me, this indicates QASUME data has a similar level of temperature dependency to Dobson. Any comments? As previously stated, I am not sure if the current QASUME algorithm retrieved effective T or used interpolated values (from radiosondes). The Pandora team seems also work on direct retrieve effective T, but had many challenges (at least, no published results or dataset yet). I would point out that Kerr 2002 also retrieved effective T, which shows pretty nice agreement with ERA reanalysis results (well, my unpublished research). It is very sad to see some knowledge in this community is not properly adapted and get lost.

Figure 7. I did not find any description of the red shading area in the paragraph. The caption here says "red bar" indicates D/Q comparison is consistent for all cross-sections. But even only eyeballing the areas, I could see some differences. Please clarify the meaning of the red shading area and provide necessary discussions. It seems this period (red shading areas) shows an opposite seasonality (decreasing with time) when compared with B/Q comparisons (increasing with time). The very strange/visible dimple in the red shading areas looks very likely due to the inaccurate effective temperature being used in the algorithm (or retrieved via the algorithm?). Please note, this feature is not shown in the B/Q comparison. Some investigation is needed. Also, please make the x-axis in Figure 7 has minor ticks. Otherwise, it is very difficult to tell the time of the observations/comparison.

Figure 4 says that when using IGQ cross-section, QASUME has a similar temperature-dependency as when using IUP or IUPA (i.e., about 0.1%/K). But, this may not be reflected in Figure 7. For example, Figure 7 shows that IUPA and ACS might have the best (lowest) relative seasonality, when compared with Brewer. Also, although when using IUPA, there is a 1% relative offset between B/Q and D/Q pairs (Fig. 7, 4[th] row), the seasonality difference is lower than the results for IUP (i.e., Fig. 7, 5[th] row, left panel ). The worries are the good agreement between Brewer and Dobson (with use IUP cross-section,) might be due to wrong reasons. To me, IUP_A might be a good choice too. Relative seasonality between two instruments is always a clear indication/signal that one of them is wrong (at some level). Please provide some comments and reasons (if possible).

L370-371: I think that starting from here, all analyses were done with the IUP cross-section (QAUME, Brewer, and Dobson). If so, please include this information in the caption of Figure 8. Another question

is more challenging. I.e., do you see different slant ozone dependency when using different cross-sections? If yes, some results could be shared in this work (e.g., with a table, or bar plots). Some simple quantification could be made, e.g., using the parameters of the fitted lines in Figure 8.

L375-377: Unfortunately, I could not agree with this. Again, there is only one true ozone, although we cannot know the truth. But, less or even more sensitivity to many different (and selected) ozone cross-sections cannot prove it is an "advantage". Well, the finding itself here is important, but I would suggest phrasing the message carefully.

L377-382: This is an important finding, i.e., constant bias when compared with Dobson (no matter which cross-section was selected). If the IUP cross-section is the future WMO standard for both Dobson and Brewer data, do we expect to see QASUME instruments will always have this 1% offset (for all sites?)? Another interesting thing is, with current results, this 1% offset is not related to the selection of cross-sections when compared with Dobson (i.e., only small changes in bias from -0.72% to -1.01% with different cross-sections). Given the standard uncertainty of only 0.8%, what are the potential sources for this large relative bias? Some further comments and discussions on this offset are welcome.

**Technical correction:**

L27: define WMO here; move the definition from L48-49 to here.

L29: change "1980's" to "1980s"

L57: please provide the temperature here.

L160: define $SO_2$ where it was mentioned the first time.

L280: the link is broken.

L575: change "standard eviation" to "standard deviation".

L354-355: please rewrite this sentence, it is a bit ambiguous.

L361: % sign is missing for the number -0.72.

**Reference**

Kerr, J. B.: New methodology for deriving total ozone and other atmospheric variables from Brewer spectrophotometer direct sun spectra, J. Geophys. Res., 107(D23), 4731, https://doi.org/10.1029/2001JD001227, 2002.

---

## Author Comment (AC1)

**Response to Reviewer #2**

We acknowledge the detailed and helpful comments of Referee #2 on our manuscript. Below you can find our specific answers to the comments in blue. We have also revised the manuscript according to the suggestions by both Reviewers. The changes are marked with red in the revised word document. For extended changes we have indicated the line-number of the revised manuscript. We have acknowledged the two anonymous reviewers in the acknowledgements.

**General Comments:**

The paper by Egli et al. presented research on deriving traceable total column ozone with the QASUME instrument. Traceability, or more specifically, a detailed uncertainty budget, is in great need for ground-based ozone observations. The methodology and analysis applied are comprehensive and solid. The instrument shows a good potential to be an independent/third-party reference to Brewer and Dobson ozone observations. However, some of the results still need more investigation or clarification. For example, the -1% offset between QASUME and Brewer or Dobson should be investigated (or, at least better described with more details), as the author claimed a 0.8% standard uncertainty. Overall, the paper is a well-written one. I recommend publishing it on AMT after addressing the following comments.

Many thanks for this appreciation of our study. We agree that the 1% bias of the traceable TCO measurements compared to Brewer and Dobson is crucial point of the publication. However, one should note that this 1% difference is well within the combined uncertainties of the TCO retrieval of Brewer, QASUME, and Dobson. In fact, a strict uncertainty analysis has so far only been performed for QASUME while it is still pending for Brewer and Dobson TOC retrievals. If we assume a very low 1% expanded uncertainty for Brewer and Dobson, covering 95% of the coverage interval, then the combined uncertainty when comparing QASUME (combined uncertainty of 1.6%) with any of those instruments would be  $sqrt(1^2+1.6^2)=1.9\%$ , which is larger than the observed bias.

TCO from QASUME is retrieved with a well described procedure. We have investigated the relevant parameters that could have an impact on the TCO retrieval of QASUME, as seen in the manuscript. For example, potential sources (e.g. Angstrom parametrization instead of linear parametrization, or different standard atmospheres) could not explain the observed bias of 1%. One can also speculate that a systematic bias could originate from imperfect Langley-plot calibrations form the Brewer and Dobson due to unknown residuals of ozone in the Langley plot. However, such speculations exceed the aim of the publication and requires more substantial scientific efforts.

We have revised the manuscript accordingly (e.g. lines 391 – 399, 405-423).

**General comment from our side:**

Due to the comment of Reviewer 1 who suggested a better guidance for the reader, we have reworded some sentence of the manuscript.

During the revision of the manuscript, we have added two corrigenda:

- 1. For the final standard retrieval, we have used the SO2 cross section HITRANS from Hermans (2019) (Line 203)
- 2. The uncertainty from solar spectra is corrected to be 0.196 % since the factor 0.5 was not included in the calculation (line 285, Table 1).

Specific comments:

L63-64: please consider changing "the four wavelengths of ... " to "the standard four wavelengths of ... ". Please note that Kerr (Kerr, 2002) developed a scanning TCO retrieval method for the Brewer, and published the work back in 2002.

Done – the proposed literature was included in line 304.

L159: If the term  $\tau$  is a function of T and p, please make it consistent. In this line, the term was referenced as  $\tau(T)$  and  $\tau(T,p)$ . Please clarify. Done

Eq. 3: Is this should be  $\tau_{AOD}$ ? (i.e., not "AOD = ... ") Done

L212-213: Some justification of this assumption is needed.

The impact of potential spectral correlations are included and discussed in the uncertainty assessment from two different solar reference spectra. Also suggested by reviewer #1 we have justified or assumption in the revised manuscript (lines 218-219, 290-292)

L227: What is the major difference between these IUP\_A and IUP cross-sections? Some description is needed.

We have added more description in the revised manuscript (lines 247 - 249).

L233: This could be a typo, i.e., "180 233 K". Please double check. *Typo – Done*

L248: The meaning of "(k = 1)" is not described (same request to the "k=2" in the later part of the paper).

We have clarified the meaning of K=1 and K=2 in the first subsection (3.1) of section 3.

L269-274: It might not be very straightforward for the reader to understand why IUP was selected (by WMO), if DBM shows less temperature dependency. Could you please provide some comments? For example, as I know, Pandora is using DBM.

The ozone absorption cross-sections of Serdyuchenko et al. 2014 were chosen by the scientific advisory group for ozone and UV of the WMO because the consistency between Brewer and Dobson TOC retrievals was considerably improved with this particular cross sections, compared to the others investigated (Redondas et al., 2014, Gröbner et al., 2021).

L277-279: I am a bit confused here. Is the effective T been retrieved or not? If it is retrieved, the results should be presented, and some description is needed.

The effective ozone temperature is not retrieved. Indeed, during the development of the algorithm, we thoroughly tried to retrieve this parameter by the least square fit algorithm, but this attempt failed. We have clarified in the revised manuscript (Lines 290 - 307).

L 326-328 and Figure 5: The right panel has y label as "Standard Deviation/Uncertainty". What is this "uncertainty"? Is this the nominal one (i.e., "arithmetic results")? But the lines in the paper described it simply as "uncertainty", not "Standard Deviation/Uncertainty". Please clarify. *We have clarified (Lines 360 - 370) and changed the figure.*

L358-359: I think this "averaged TCO" should be "averaged relative difference (of TCO)". *We agree and have clarified in the revised manuscript accordingly.*

L362-364: The results here are very interesting and important. I do not want to be hypercritical. However, this information/claim might be very misleading too. First, lower variability for the Dobson-QASUME (D/Q) pair only can prove they (these two instruments) have a similar response (to the set of ozone cross-sections studied here). Second, the results depend on which cross-section is used. If we only select IGQ and IUP, the differences are very small (I doubt if the difference is statistically significant). Is this sensitivity due to instrument or due to cross-sections, or both? Anyway, the point is there is only one true ozone value. For any instrument, being sensitive to a "non-ideal" ozone cross-section might not be a bad thing. I would suggest rephrasing this part as "Brewer is more sensitive to some of the ozone absorption cross-sections (e.g., … ) than … ".

Indeed this is an important comment. We fully agree with the reviewer that the discussion on the results need to be formulated carefully. It basically cannot be prooven what causes the sensitivity on the cross-section for Brewer, Dobson and QASUME. We have rephrased the section more conservatively (Lines 395 - 399)

Figure 7. Based on previous publications, Brewer has a remarkably low temperature-dependency (e.g., Kerr 2002), or at least "theoretically" better than Dobson (even with Bass&Paur, or IGQ here). However, the top panel in Figure 7 show that QASUME agrees better with Dobson; it shows a stronger seasonal structure when compared with Brewer. To me, this indicates QASUME data has a similar level of temperature dependency to Dobson. Any comments? *We have stated the similar temperature dependency of Dobson and QASUME more clearly in the revised manuscript (line 292)*

As previously stated, I am not sure if the current QASUME algorithm retrieved effective T or used interpolated values (from radiosondes). The Pandora team seems also work on direct retrieve effective T, but had many challenges (at least, no published results or dataset yet). I would point out that Kerr 2002 also retrieved effective T, which shows pretty nice agreement with ERA reanalysis results (well, my unpublished research). It is very sad to see some knowledge in this community is not properly adapted and get lost.

As clarified earlier, the effective temperature is not retrieved by the QASUME algorithm. We have used the linearly interpolated values from Gröbner et al. 2021. In the revised manuscript, we have acknowledged the findings of Kerr 2002, who successfully retrieved effective ozone temperature from the Brewer. This retrieval from Brewer could not be done with other ground based TCO instruments so far and should be highlighted here. We also see the importance to retrieve effective temperature, however we made substantial effort to do this in the presented study, but we were not successful (see lines 398 – 308 and 175- 179)

Figure 7. I did not find any description of the red shading area in the paragraph. The caption here says "red bar" indicates D/Q comparison is consistent for all cross-sections. But even only eyeballing the areas, I could see some differences. Please clarify the meaning of the red shading area and provide necessary discussions.

The red shaded are should only highlight the offset numbers and not the corresponding data points in the shaded area. We apologize for this ambiguity. The red shaded is removed. The values of the offset are discussed in the manuscript.

It seems this period (red shading areas) shows an opposite seasonality (decreasing with time) when compared with B/Q comparisons (increasing with time). The very strange/visible dimple in the red shading areas looks very likely due to the inaccurate effective temperature being used in the

algorithm (or retrieved via the algorithm?). Please note, this feature is not shown in the B/Q comparison. Some investigation is needed.

We have used the same effective temperature from the balloon soundings for the Brewer, the Dobson and the QASUME retrieval. We have clarified this in the revised manuscript (line 383)

Also, please make the x-axis in Figure 7 has minor ticks. Otherwise, it is very difficult to tell the time of the observations/comparison.

We have adopted the figure accordingly.

Figure 4 says that when using IGQ cross-section, QASUME has a similar temperature-dependency as when using IUP or IUPA (i.e., about 0.1%/K). But, this may not be reflected in Figure 7. For example, Figure 7 shows that IUPA and ACS might have the best (lowest) relative seasonality, when compared with Brewer. Also, although when using IUPA, there is a 1% relative offset between B/Q and D/Q pairs (Fig. 7, 4th row), the seasonality difference is lower than the results for IUP (i.e., Fig. 7, 5th row, left panel ). The worries are the good agreement between Brewer and Dobson (with use IUP cross-section,) might be due to wrong reasons. To me, IUP\_A might be a good choice too. Relative seasonality between two instruments is always a clear indication/signal that one of them is wrong (at some level). Please provide some comments and reasons (if possible).

As shown in Gröbner et al., 2021, the Brewer is more sensitive to changing ozone absorption crosssections than a Dobson or QASUME (See Figure 5 in Gröbner et al., and figure 2 in this manuscript). Thus, small errors in the cross-sections will have a larger impact on the Brewer than either QASUME or Dobson. This is also consistent with the data shown in Figure 7, where changes in ozone absorption cross-sections show larger variability when comparing QASUME to the Brewer than to the Dobson. The conclusion that can be drawn from Figure 7 is that none of the ozone absorption cross-sections investigated in this study produce perfect consistent results between the three instruments, so small errors (spectral as well as absolute) remain in all these cross-sections. The selection of the most consistent cross-section based on the lowest bias and smallest seasonal variability is the IUP crosssection, even though future work is clearly needed to resolve the still observed discrepancies between these three instruments.

L370-371: I think that starting from here, all analyses were done with the IUP cross-section (QAUME, Brewer, and Dobson). If so, please include this information in the caption of Figure 8. Indeed the slant path analysis is done with IUP cross section only since we have chosen this cross section for the standard algorithm. We have added this information in the caption.

Another question is more challenging. I.e., do you see different slant ozone dependency when using different cross-sections? If yes, some results could be shared in this work (e.g., with a table, or bar plots). Some simple quantification could be made, e.g., using the parameters of the fitted lines in Figure 8.

Many thanks for this suggestion. We have additionally investigated the slant path dependency (differences of the linear fit at 300 and 1200 DU slant path) of the individual cross sections as follows:

| Brewer 156 | Dobson 101                                               |
|------------|----------------------------------------------------------|
| 0.29%      | 0.418 %                                                  |
| 0.07%      | 0.03%                                                    |
| 0.12%      | 0.40%                                                    |
| 0.09 %     | 0.43%                                                    |
| 0.02%      | 0.43%                                                    |
|            | Brewer 156
0.29%
0.07%
0.12%
0.09 %
0.02% |

The results show that the slant path dependency is mostly insensitive to the selected cross section for the Brewer as well as for the Dobson. Except for IGQ applied on the Brewer shows significant higher slant path dependency, which is attributed to the seasonal variation when using IGQ. Since the solar zenith angle is correlated with season and corresponding effective temperature, the seasonal variation in temperature is also reflected in the slant path dependency. This detail is not further explained in the revised manuscript.

*However, we have stated the finding of mostly insensitivity of slant path dependency on cross sections in the revised manuscript (Line 405 - 407), which is of interest for the reader.*

L375-377: Unfortunately, I could not agree with this. Again, there is only one true ozone, although we cannot know the truth. But, less or even more sensitivity to many different (and selected) ozone cross-sections cannot prove it is an "advantage". Well, the finding itself here is important, but I would suggest phrasing the message carefully.

We agree that the statement of the "advantage" is inappropriate based on the sensitivity analysis regarding cross section. We have reformulated the statement more conservative (304-309).

L377-382: This is an important finding, i.e., constant bias when compared with Dobson (no matter which cross-section was selected). If the IUP cross-section is the future WMO standard for both Dobson and Brewer data, do we expect to see QASUME instruments will always have this 1% offset (for all sites?)? Another interesting thing is, with current results, this 1% offset is not related to the selection of cross-sections when compared with Dobson (i.e., only small changes in bias from -0.72% to -1.01% with different cross-sections). Given the standard uncertainty of only 0.8%, what are the potential sources for this large relative bias? Some further comments and discussions on this offset are welcome.

See answer of the general comment. Preliminary comparison results at two other locations e.g., at the Observatory in Izana, Teneriffe, Spain in 2016 and in El Arenosillo 2019 in Southern Spain showed also about 1% bias compared to Brewer. The final approved results will be published in a WMO report in 2022.

We have revised the manuscript accordingly (Lines 413 -419).

L27: define WMO here; move the definition from L48-49 to here. Done L29: change "1980's" to "1980s" Done L57: please provide the temperature here. Done L160: define SO2 where it was mentioned the first time. Done L280: the link is broken. Sone - The link is check and it is working. L575: change "standard eviation" to "standard deviation". Done L354-355: please rewrite this sentence, it is a bit ambiguous. Done L361: % sign is missing for the number -0.72. Done

---

## Author Comment (AC2)

**Response to Reviewer #1**

Many thanks for the helpful work of Referee #1 on our manuscript. Below you can find our specific answers to the comments in blue. We have also revised the manuscript according to the suggestions by both Reviewers. The changes are marked with red in the word document. For extended changes we have indicated the line-number of the revised manuscript. We have acknowledged the two anonymous reviewers in the acknowledgements.

**General Comments**

The authors present a retrieval for the measurement of total column ozone based on the "QASUME" reference spectroradiometer, based on absolute measurements of the spectral irradiance which are traceable to SI. This general concept is a break from the traditional notion of "traceability" in the ozone community which has historically always meant traceability to an artefact, namely Dobson 83 or the Toronto Brewer triad.

The manuscript is very relevant and suitable for publication in AMT with minor revisions. Most of my specific comments below are essentially just requests for a bit more guidance for the reader.

Due to this comment and due to the comments of Reviewer 2, we have re-worded some sentence of the manuscript, for better guidance for the reader.

*During the revision of the manuscript, we have added two corrigenda:*

- 1. For the final standard retrieval, we have used the SO2 cross section HITRANS from Hermans (2019) (Line 203)
- 2. The uncertainty from solar spectra is corrected to be 0.196 % since the factor 0.5 was not included in the calculation (line 285, Table 1).

My one overall comment is that I would appreciate some more introductory motivation and discussion for the concept of a "traceable" retrieval, as I suspect this idea will not be very familiar to many of the readers of AMT (in contrast to the idea of a traceable measurement in a laboratory setting, or the concept of an uncertainty budget for an atmospheric retrieval, both of which are widely understood).

What is the motivation for the approach taken here in the manuscript, compared to the traditional Dobson and Brewer concept of traceability? (You do already give two advantages in lines 16-17 I note).

We appreciate this positive review of our manuscript. Regarding "Traceability" we have introduced the motivation in the abstract and now included some more introduction of the motivation of the work in the revised manuscript (e.g. Lines 66 – 73, Abstract and Conclusion).

Some specific points I would like to see discussed are as follows:

Is the definition of "traceability" of a retrieval you give (line 73), in common use?

We clarify that we have used our own definition of "traceability" for retrieval with a model. To our knowledge there is not a common definition of traceability regarding retrieval models. We have clarified in the revised manuscript. (Lines 73).

Additional data are required in the retrieval, for example profiles of ozone and temperature measured by ozonesondes. However these measurements are not traceable, unlike those from QASUME – what is the impact of that?

Many thanks for this hint. Indeed, some input parameters are not traceable. In order to take this into account, we have used a very conservative value for the estimated uncertainty of these components. The impact of the uncertainty of the effective ozone temperature from sondes and reanalysis data is investigated and stated in more detail in the revised manuscript (lines 304 - 308).

The uncertainty is based specifically on conditions in Davos. However, the text states the instrument has been used at 33 different locations around the world. Does this mean the traceability of the ozone retrieval does not apply at these locations?

We have highlighted that the results of his study are based on the specific location at Davos. We also clarify, that QASUME is used at the mentioned 33 station for global UV irradiance and not for direct spectral solar irradiance measurements to retrieve total column ozone. At two specific locations (Teneriffe and El Arenosillo, Spain) QASUME measured direct solar irradiance and retrieved TCO. The results will be published in a WMO report in 2022. Preliminary results compared to Brewer showed similar results as in this study. We now mention the two other sites in the revised manuscript (also according to the suggestion of Reviewer 2 (lines 413 - 417).

Even though you have an established climatology it is always possible for the atmosphere at any one moment to be in a very unusual state, or even in an unprecedented state – how does your approach cope with that? Can you still say "accounting for all possible uncertainties"? (Line 19)

Indeed, this statement is a bit exaggerated. We agree with the Reviewer that might be some uncertainties which are not considered in the study. We have removed "for all possible uncertainties".

**Specific Comments**

Line 28 The website is useful but you need to also give a formal citation, for example to the most recent assessment.

**Done**

Line 32 "Variations of the solar constant" isn't the main point here.

*Done – we have removed this part*

Line 45 perhaps "with the then state-of-the-art"

**Done**

Line 55 What do you mean by "fundamental" here? You then go on to say most of the difference can be accounted for if well-understood issues are taken into account.

We have removed "fundamental" and added "biases" instead of "uncertainties"

Lines 58-60 I think it would be better to list a wider range of instruments here that have also used a similar spectral range even if not in the same geometry. For example, NDACC UV spectroradiometers have been making similar measurements for decades (although global, not direct) from which it is possible to retrieve total ozone. Similarly there are the DOAS and MAX-DOAS instruments which are nowadays very widely used (using zenith rather than direct-sun measurements of course).

**Many thanks for your suggestion. However, we in this study we would like to focus on TCO instruments measuring direct irradiance only.**

Line 64 Actually, I would have "expected" the opposite. Measuring the full spectrum gives you many more data points but, each point does not necessarily contain any additional independent information and, speaking very generally, a ratio is usually able to be measured much more precisely than an absolute quantity.

**We agree that our statement is rather speculative. We have removed the sentence.**

Line 72 This definition of "traceability" (for a retrieval) is not the same as the definition of traceability for an instrument. You need to explain to the reader whether this definition is your own or is generally used.

*Indeed, we have used our own definition of "traceability" for retrieval with a model. We have clarified in the revised manuscript (lines 73).*

Line 111 Is the airmass for ozone?

**Yes. Done**

Line 119 Why is "traceable" in quotation marks here – is the implication that you are using the word in a particular way?

**Since we have defined "traceable" earlier we have removed the quotation mark.**

Lines 147-151 It seems a bit odd to me that you would show the plot of residuals before giving any details of what you are fitting. Normally in a paper this would be the other way around. Perhaps you could let the reader know the details are coming.

**We have clarified this issue.**

Line 152 Saying the residuals are "flat" seems a bit optimistic to me, there is some possible structure there, apart from the high frequency variation

We have formulated the sentence more conservatively (line 151)

**Lines 152-154 Is it possible to explain this more clearly?**

We have reformulated the sentence (line 153 - 154)

Line 162 Using afglus can only be a starting approximation for a specific location and season etc

*Recalculation TCO with other standard atmospheres than afglus revealed neglectable changes of TCO in Davos. We have revised the manuscript accordingly (line 191).*

Line 174 Using values for that particular day, or an average, or something else? Would this introduce an additional uncertainty?

As in Gröbner et al 2021 and specified in the manuscript we have used interpolated values from soundings every 2 or 3 days (lines 175 - 178).

Lines 171-174 As mentioned in my general comments, the use of outside information like the Payerne ozonesondes provokes a number of questions. Ozonesondes and radiosondes have their own issues of course and are not traceable to SI. Do you know how representative Payerne is for the vertical structure of ozone over Davos?

Gröbner et al. 2021 compared the effective temperature from Payerne with ECMWF reanalysis data from Davos. The differences between the two datasets was defined as the uncertainty of effective temperature in the manuscript in section 3.4. We have added that the effective temperature is linearly interpolated for the missing days and smoothed with a 10 days tunning mean as in Gröbner et al. 2021 (lines 174 - 177).

Lines 175-181 I think you should mention the limitations of this assumption for the AOD.

A more appropriate parametrisation would use the Angstroem approximation (AOD=beta . lambda^alpha, with lambda the wavelength in micrometer. The tests we have performed have shown that the retrieval of TOC is not affected by using either approach (linear parametrisation of AOD or using the Ansgtroem law). This is essentially due to the short spectral range of 45 nm that is used for the retrieval algorithm, between 305 nm and 350 nm, where a linear or Angstroem AOD fitting function does not show any significant deviations. Clearly, for a larger spectral interval, the use of the Angstroem fitting function would be more appropriate (lines 184 – 185).

Lines 185 You should give at least some details of your assumptions for SO2, and how good you expect them to be, particularly as SO2 is likely to be highly variable in time and space. Lines 190-199 How do you know there will not be other non-negligible absorbers within this wavelength range?

Also in Gröbner et al. 2021 it can be found that: "the main atmospheric absorber in the measured wavelength band is ozone. Even though sulfur dioxide and nitrogen dioxide also absorb in this wavelength range, their amount in the atmosphere above Arosa and Davos is so small that it can be neglected here". We have added this information in the revised manuscript (lines 188 – 189).

Line 206 Is the "overall uncertainty of 0.91%" for every wavelength?

The uncertainty is not given for every wavelength in this region of 305 – 345 nm. at wavelengths shorter than 305 nm the uncertainty is slightly larger. Therefore, we use 0.91% for the entire wavelength band.

Line 213 Shouldn't you assume the more conservative limit of 0.72% ? Is there a physical consideration here?

Effects of substantial spectral correlations can be detected by the differences of QASUMEFTS (which is derived from QASUME) and an independent solar spectrum (TSIS). The impact of potential spectral correlations is included and discussed in the uncertainty assessment from two different solar reference spectra. (lines 191 - 193). Therefore "unfavorable correlation" from the measurement uncertainty is a even more conservative assumption than "no correlation". Finally, there are no correlations stated in

the uncertainty assessment of QASUME, the cross section or the solar spectrum. We have clarified in the revised manuscript. (e.g. lines 214, 218 – 219, 289 – 291).

Lines 215-216 Wouldn't the better approach here be to consider the uncertainties of the laboratorymeasured ozone cross-section, (which I understand was one of the original motivations for the ATMOZ project) and propagate them through to the resulting total ozone value?

Lines 245-248 I don't follow the reasoning here. The ozone cross section at a specific temperature and wavelength has a true value which can in principle be measured, and we hope is getting more accurately measured as laboratory techniques improve. Why even consider the older Bass & Paur values at all?

This is a good point. Indeed, we have calculated the impact of the uncertainty given by the cross section as random (uncorrelated) noise to the cross section. However, due to the convolution of the cross section the noise was reduced, and the resulting uncertainty was less than 0.06%. To obtain a more realistic value we decided to estimate the uncertainty by comparing TCO from different cross section, assuming that all cross sections may be measured with best possible technology. Since Bass & Paur is still in use for brewer and Dobson we have also included this cross section.

We have clarified this in the revised manuscript (lines 226 - 231).

Line 240 This is quite confusing. Is there a difference between IUP and IUP\_A and IUPA?

We have clarified this in the revised manuscript. We now use IUPA instead of IUP\_A through the entire manuscript.

Lines 250-266 Again I am struggling with the reasoning. Doesn't the TSIS come with an uncertainty, which you could propagate through to the total ozone value?

According to the same argument as when considering the noise to the cross-section (uncertainty less than 0.06% due to convolution) we have chosen a second ETS to assess the overall uncertainty. We have added this argument in the revised manuscript (lines 272 – 277).

Lines 269-285 Is the standard deviation really enough, do you know whether extreme values are properly represented in the uncertainty? Also do you need to take into account the uncertainty in the mean value of 225.2 K caused by radiosonde errors or bias?

We assume that the measurement uncertainty of the radiosondes is far below 2.5K. However, the 2.5K from the comparison of soundings and reanalysis data seems to cover realistically the overall uncertainty (lines 305 – 308).

Lines 309-310 It is not clear to me what happened to the aerosol and SO2 and their effect on the uncertainty?

Aerosols and SO2 are used as fitting parameters. Their effect is covered by the computational uncertainty from the least square fit algorithm.

Line 323 Do you use a normal distribution for the random values?

*The type of the distribution is indicated in Table 1 second column. We have indicated this in the revised manuscript (line 356).*

**Technical comments**

Line 29 Remove the apostrophe

**Done**

Line 32 "surface of the Earth" or "Earth's surface"

**Done**

Line 40 "have" should be "has"

**Done**

Line 41 "to [form] a global network"

**Done**

Line 88 "has been" operated

**Done**

Line 91 Replace "chapter" with "section"

**Done**

Line 99 "enables" rather than "ensures", or perhaps "ensures outdoor measurements are able to be made"

**Done**

Line 103 Replace "timeout" with "missing "

**Done**

Line 109 Insert "to" after the word "equal"

**Done**

Line 109 "a maximum of 4.5 minutes"

**Done**

Line 112 "described" not "describes"

**Done**

Line 137 "et al."

**Done**

Line 149 "surface of the earth"

Done

Line 151 You don't mean "exemplarily" here

**Done**

Line 182 You need to re-word this, at present it reads as if Lord Rayleigh is personally scattering the photons around!!

**Done**

Line 203 Delete "well", otherwise it sounds like the authors are complimenting themselves!

**Done**

Line 207-209 I can't quite follow the meaning of this sentence, please re-word to make it clearer

We have reworded the sentence

Line 324 Apostrophe should be a comma.

Done

Line 330 Apostrophe should be a comma.

Done

Line 334 "negligibly" instead of "neglectable"

Done

Line 336 "proves" not "proofs"

This is removed due to the comment of Reviewer 2.

---

## Author Response (AR2)

**Response to the Editor Mark Weber**

Dear Mark

We appreciate very much for your additional effort to improve our manuscript. Please find below our answers to your comments and suggestions in blue.

**Your Comments**

Dear Luca and Julian,
responses to the reviewer's are adequate, but I still have a few minor issues that should be resolved.

First of all the ATMOZ ozone cross-sections has been put into a data repository and are citable: Gorshelev, Victor, Weber, Mark, & Burrows, John P. (2017). ATMOZ Gorshelev Huggins Ozone Band Absorption Cross-Section (1.0) [Data set]. Zenodo. https://doi.org/10.5281/zenodo.5847189.

*We have added this reference in the revised manuscript (line 248 and 499-501)*

I am not so happy with the abbreviations IUP and IUPA for our x-section data. I would prefer SG14 for IUP and G17 for IUPA. It would be nice to change these names.

*Many thanks for this clarification. We have clarified your abbreviation in the revised manuscript and changed all figures accordingly. IUP is labelled now as SG14 and IUPA is now G17. The changes are made thorough the entire manuscript. We have also clarified that the in Gröbner et al a different nomenclature is used. Line 59.*

Here are some further points:

l. 188: add "surface" (surface pressure)

*Done (line 188)*

l. 191 "negelectable" --> negligible

*Done (line 190)*

l. 307 if --> is

*Done (line 310)*

l. 314 within --> with (or do you mean no seasonal dependence on the correlation? Please rephrase.

*Done (line 318)*

l. 333-338: Does the retrieval uncertainty not include uncertainties from input data used in the retrieval, e.g ETS, eff temp., and so on, so some double counting occurs here?

*We assume that the uncertainty from the fitting originates from short term atmospheric perturbations such as moving clouds, cirrus clouds or other atmospheric effects (stated in line 338 - 339). ETS and cross sections are constant for the entire retrieval and the effective temperature is constant over a day. Therefore, these uncertainties are not double counted. Evenmore, we have used this parameter as a criterion to select unfeasible TCO retrievals, with disturbed direct solar irradiance, which corresponds to own observations at some specific days.*

l. 400: "which might lead to a lower sensitivity ..." am not sure if this holds. The satallite retrievals using effective temperatures show less seasonal variations in the differences to the Brewers than to Dobsons. This makes sense as Brewers have negligible temperature sensitivity compared to Dobsons (Kerr 2002). As I understand QUASUME does not retrieve effective temperatures like the Dobsons, so it seems that this could explain the higher similarity. Can you comment on this?

*Since we have included the effective ozone temperature as input for the Dobson and the Quasume retrieval, the effect of effective temperature should not affect the intercomparison. However, we agree that our statement is rather speculative. Therefore, we have removed this sentence in the revised manuscript (line 402 – 403)*

l. 400: as for --> and

*Done (line 403)*

Best wishes,
Mark